# Cultivating Healthier Habits: The Impact of Workplace Teaching Kitchens on Employee Food Literacy

**DOI:** 10.3390/nu16060865

**Published:** 2024-03-16

**Authors:** Richard Daker, Ghislaine Challamel, Chavanne Hanson, Jane Upritchard

**Affiliations:** 1Restaurant Associates (Compass Group), Charlotte, NC 28217, USA; richdaker@gmail.com (R.D.); ghislaine.challamel@gmail.com (G.C.); 2Google LLC, Mountain View, CA 94043, USA; chavannehanson@google.com; 3Bon Appetite Management Company (Compass Group), Charlotte, NC 28217, USA

**Keywords:** employees, teaching kitchens, food literacy, nutrition education, cooking, wellbeing

## Abstract

This research explores the impact of workplace teaching kitchen cooking classes on participants’ food literacy and identifies key predictors of employee engagement. Aligning with the existing literature, we demonstrate that a workplace teaching kitchen program, with hands-on cooking classes, effectively enhances food skills and intrinsic motivation—core aspects of food literacy. Importantly, our results reveal that even a single class can have a measurable impact. Teaching kitchens can successfully engage employees, particularly those with low food skills, showcasing their broad appeal beyond individuals already engaged in wellness or seeking social connection. Awareness emerges as the most influential predictor of participation, emphasizing the crucial role of marketing. Virtual classes prove as effective as onsite ones, offering the potential to increase access for employees. Recognizing employee wellness as a strategic opportunity for employers and a sought-after benefit for top talent, we underscore the importance of practical nutrition education to support individuals in shifting food choices within lifestyle constraints. Workplace teaching kitchens emerge as an effective and scalable solution to address this need. Future research should prioritize exploring the lasting impacts of teaching kitchen education on employee eating habits and health, contributing to ongoing strategy refinement.

## 1. Introduction

A healthy workforce provides a strategic and competitive advantage for employers [1,2]. Unhealthy employees typically incur higher healthcare costs and increased productivity losses [2,3]. Prioritizing health and safety not only enhances employee engagement and commitment to organizational success [4,5] but also positions a company as a desirable employer, attracting and retaining top talent [6]. These employer benefits underscore the need to identify effective strategies for addressing employee wellbeing.

Globally, the health of the working-age population is declining, with rising rates of obesity, diabetes, heart disease, and certain cancers [7,8]. Poor diet is a leading risk factor for these diseases, especially diets high in sodium and red meat, and low in whole grains, fruits, legumes, dietary fiber, and vegetables [7]. Conversely, a healthy diet protects against noncommunicable chronic diseases, including diabetes, heart disease, stroke, cancer, and malnutrition [9]. Adhering long-term to healthy dietary patterns, regular physical activity, and maintaining an acceptable weight can prevent the majority of noncommunicable chronic diseases [10,11] as well as improve outcomes in those at risk or diagnosed with a chronic disease [12]. Good nutrition and healthy eating habits are associated with maintaining health, and emerging evidence shows benefits to workforce productivity, through improved cognitive function, mental focus and increased energy levels [13]. 

Healthy dietary practices involve consuming a variety of fruits, vegetables, legumes, nuts, and whole grains, with limited sugars and fats and balancing portion sizes with energy requirements [14,15,16,17]. However, increased urbanization and lifestyle changes have led to a higher consumption of ultra-processed foods and ready-made meals [18,19,20], associated with energy-dense, high-sodium, high-sugar, harmful fat-laden diets, low in fiber, with minimal or no whole foods, prone to overconsumption [18,21]. Barriers to home meal preparation include both real and perceived time constraints, perceived affordability, and a dislike for cooking [22,23,24,25,26]. 

Achieving dietary change requires more than theoretical knowledge; it requires access to wholesome foods and the skills to transform them into nutritious meals [27]. Cooking skills are instrumental in reducing reliance on processed foods and ready-made meals [28,29,30,31], yet these skills have been declining for decades [26,31,32,33,34], potentially lost across multiple generations in urbanized populations [35]. A comprehensive review of thirty-four multidisciplinary nutrition education interventions revealed that incorporating a cooking component, particularly through hands-on cooking classes, consistently led to improved participant food choices and diets [36]. Recent research further validates this conclusion [37,38,39,40]. However, creating delicious, nutritious meals within the constraints of an individual’s lifestyle demands more than just the mechanical skills of preparing food and cooking [41]. The concept of food literacy has emerged as a useful framework to encompass the various factors influencing an individual’s ability to acquire cooking skills and translate them into practice within the context of their social, physical, and economic environments [42,43,44,45]. 

Teaching kitchens seamlessly blend nutrition education with hands-on cooking experiences within a social learning environment [46]. These kitchens go beyond merely imparting culinary skills, they cultivate cooking as a habitual practice, empowering individuals to overcome barriers and adapt to daily life conditions, thereby enhancing food literacy. Teaching kitchen programs deployed in clinical settings show promising results on management of cardiovascular and diabetic risk factors [47,48,49], presumably through sustained changes to diet [47]. Outside of clinical settings, teaching kitchen programs appear to be effective at shifting diets among a variety of population groups, employees, college students, families, and adults with low food literacy [39,40,50,51,52,53].

Building on the promising outcomes of a prior workplace feasibility pilot study [50], our research seeks to assess the impact of workplace teaching kitchens on the development of food literacy in employees. By focusing on markers of food literacy, we aim to expand the current understanding of the role of teaching kitchens in the early stages of behavior change. This research aligns with the evolving landscape of workplace wellness initiatives aiming to shift food choices and build healthy sustainable eating habits. This study’s primary objectives are to (1) evaluate the impact of a workplace teaching kitchen program on measures of food literacy in employees and (2) identify the key predictors of employee engagement with the teaching kitchen program.

## 2. Materials and Methods

### 2.1. Study 1

Study 1 is a single-group prospective cohort study that relies on institutional data collected as part of continuous monitoring of outcomes of an established teaching kitchen program at an American multinational technology company. The primary goal of this study was to understand whether class attendees grew in key metrics of food literacy after taking classes.

#### 2.1.1. Methods

Study participants were employees at an American multinational technology company who completed classes from 1 March 2023 to 17 August 2023. Data collection included pre- and post-class surveys. All class attendees were required to complete a pre-class survey as a prerequisite before taking their first class. In this period, a total of 9665 employees completed the mandatory pre-class survey. After each class attendees took during this period, they were offered the opportunity to complete an optional post-class survey. During the study period, 720 employees completed at least one optional post-class survey. Demographic information was not collected to adhere to employee privacy policies. Externally published data on the company employees reported 66.8% of employees have a bachelor’s degree and 14.5% have a master’s degree; 59% are aged between 20 and 30 years and 19% between 30 and 40 years; 68.5% of employees are male; 49.7% are White, 18.2% are Asian, 18.28% are Hispanic or Latino, 8.2% are Black or African American (Google Diversity Annual Report 2023, Available online: https://about.google/belonging/diversity-annual-report/2023/ (accessed on 8 February 2024)).

All survey data collection for Study 1 was carried out through an online platform (Appointy Software Inc., Lewes, DE, USA, https://www.appointy.com, accessed on 8 February 2024). This platform supports class schedule, enrollment, and survey management. All study procedures were approved by Google Ethics & Compliance, and participants provided informed written consent in accordance with Google Employee Privacy Policy.

Upon signing up for their first class during the study period, attendees received a link to the pre-class survey. The survey included a series of questions aiming to measure attitudes toward different aspects of food literacy (see Section 2.1.2 below for details). Shortly after completing each class, during the study period, attendees received an email with a link to an optional post-class survey. This survey included a brief section where attendees could give feedback on the specific class they attended, followed by an opt-in survey where they again answered the same food literacy questions included in the pre-class survey. By offering the food literacy survey questions as part of an opt-in section, the reliability of the answers was improved without deterring participants from giving specific feedback on the class they had attended. Attendees were invited to complete post-class surveys for each class they attended during the study period. The primary goal of the post-class survey analysis was to enable understanding about how attendees changed on measures of food literacy after having taken classes; for each participant who completed post-class surveys, the most recent (last) post-class survey was used in analysis.

#### 2.1.2. Materials

The intervention for this research was consumption of one or more classes from the established workplace teaching kitchen program at a multinational tech company. Teaching kitchens are co-located in company offices available in 39 cities worldwide and globally through a virtual online platform spanning all 54 countries the tech company has employees. The program offered is the same whether accessed through an onsite teaching kitchen or online, including the class format, duration, audiovisual tools, curriculum, syllabus, and teaching pedagogy. The latter are described in more detail in the next paragraph. The teaching kitchen spaces and program are based on the concept described by Eisenberg and colleagues [46].

The teaching kitchen program uses a centralized design approach that is customized to local ingredients and cultural preferences by the local teachers. The program is grounded in a behavior-centered design strategy that incorporates key drivers of behavior change, which provides participants the opportunity to perform and practice the desired behavior, fostering intrinsic motivation and building capability for successful action [54]. In this case, the desired behavior is to select and cook with wholesome, unprocessed foods to create delicious nutritious meals and snacks personalized for the participant’s preferences and lifestyle. The teaching kitchen program curriculum integrates principles advocating for healthy and sustainable eating, aligning with the Culinary Institute of America’s Healthy & Sustainable Menu Principles [55] and contemporary sustainable dietary guidelines [11,14,15,16,17]. The pedagogical structure is based on a food literacy framework [44,45,56,57]. Sessions focus on planning, ingredient selection, preparation, cooking, serving, and eating, emphasizing skill-building, adaptability, and decision-making applicable in diverse food environments [27,42]. Fundamental skills such as mise en place, knife proficiency, ingredient ratios, and flavor balancing are integral aspects of each class. The syllabus is tailored to local interests, culture, and environment, covering seasonal and local foods, global cuisines, cultural favorites, and celebratory dishes. Classes are led by trained culinary professionals proficient in the program’s pedagogy. They employ participant engagement strategies following the principles of social cognitive theory, fostering a hands-on group learning environment that integrates personal factors, environmental influences, and behavior [58].

Participants completed the pre- and post-class surveys to evaluate self-reported change in measures of food literacy. These measures of food literacy were grouped in four domains: importance of learning experience, ease and pleasure of cooking, food skills, and confidence in the kitchen. The survey questions were adapted from the International Food Literacy Survey [56] with some more specific cooking practices questions from the Cooking and Food Provisioning Action Scale Survey [59]. Both surveys had been previously validated [56,59]. The order of questions within each survey set was randomized. For a list of all survey items, see the Appendix A.

Importance of learning experience refers to the degree to which people value home-cooked, healthy, sustainable eating. This construct was measured with 7 items. Examples include “It is important to eat home-cooked food” and “I want to know more about how foods can improve my wellbeing”. Participants were asked to indicate the extent to which they agreed with the statement in each item on a 5-point scale from strongly disagree (0) to strongly agree (4). Importance of learning experience scores for each participant were computed by taking the average of the 7 items, and possible scores range from 0 to 4. Cronbach’s α (using the 9965 attendees who completed the pre-class survey) for this measure was 0.91.

Ease and pleasure of cooking refers to the degree to which participants find that cooking is simple and enjoyable. This construct was measured with 5 items. Examples include “I enjoy cooking” and “I find it simple to cook food”. Participants were asked to indicate the extent to which they agreed with the statement in each item on a 5-point scale from strongly disagree (0) to strongly agree (4). Ease and pleasure of cooking scores for each participant were computed by taking the average of the 5 items, and possible scores range from 0 to 4. Cronbach’s α for this measure was 0.87.

Food skills refers to self-rated ability on several key cooking skills. Participants were instructed to “Use the slider to tell us how you would rate your food skills today” and were shown a series of 9 items. Example items include “Plan meals ahead of time”, “Knife skills (chopping, slicing, dicing…)”, and “adapt recipes”. Ratings were made on a slider from a frowny face (1) to a smiley face (5), and only integers were selectable. Food skills scores for each participant were computed by taking the average of the 9 items. We subtracted 1 from this average to arrive at a possible range of food skills scores from 0 to 4, which is the same range as the other food literacy measures. Cronbach’s α for this measure was 0.93.

Confidence in the kitchen refers to the extent to which participants feel confident about their ability to cook and eat well. Participants are asked to “Indicate the extent to which you FEEL CONFIDENT about performing each of the following activities”. The three items are as follows: “Preparing a tasty meal using mainly fresh and whole food ingredients”, “Eating the recommended amount of vegetables and fruit each day”, and “Planning a balanced meal”. Responses were made on a scale from “Not at all confident” (0) to “Very confident” (4). Confidence in the kitchen scores for each participant were computed by taking the average of these 3 items, and possible scores range from 0 to 4. Cronbach’s α for this measure was 0.90.

Institutional data were collected on the specific classes each participant took, whether they participated in an onsite or virtual class, location, and employee function.

### 2.2. Study 2

This study focuses on identifying factors that predict participation in cooking classes and includes a comparison group of individuals who have not taken classes to provide a control context for Study 1’s findings. Exploring predictors of both class attendance and non-attendance is crucial for teaching kitchen programs to enhance engagement effectively within a workplace population. While the earlier analyses reveal changes in food literacy following classes, they lack a comparison with individuals who did not participate. This absence of a comparison group raises the possibility that individuals naturally develop improved food skills over time, irrespective of class participation. To address these gaps in understanding, we conducted an additional study.

#### 2.2.1. Methods

Study 2 participants were employees at the same American multinational technology company as those from Study 1. Rather than relying on institutional pre- and post-class survey data of class attendees as we did in Study 1, for Study 2, we collected data from a broader sample of employees. 

Recruitment occurred via an internal online message board that employees can visit to find out about events and training opportunities. Study opportunities and recruitment efforts were active in over 50 countries around the world. Participants were told that they would complete a survey to share their thoughts on food. Participation in the survey involved taking a survey at two timepoints—one survey was to be completed when they signed up for the study, and another was to be sent to them 6 weeks after completing the first survey. 

A total of 272 participants completed the first survey, and a total of 156 participants also completed the second survey. Demographic questions were optional, but of the 89 who indicated their gender, 42% were male, 57% were female, and 1% were non-binary. Additionally, of the 119 who indicated their age range, 29% were between the ages of 20 and 30 years, 42% were between the ages of 31 and 40 years, and 29% were above the age of 40 years.

All survey data for Study 2 were collected and managed through Qualtrics (Qualtrics LLC., Provo, UT, USA, https://www.qualtrics.com, accessed on 31 September 2023). All study procedures were approved by Google Ethics & Compliance, and participants provided informed written consent in accordance with Google Employee Privacy Policy. The survey included the same items as those in Study 1 with some additional items added (see Section 2.2.2 below). After completing the pre-survey, participants were sent an email 6 weeks later to complete the post-survey. 

#### 2.2.2. Materials

Participants completed two surveys. This included each of the food literacy surveys from Study 1: importance of the learning experience, ease and pleasure of cooking, food skills, and confidence in the kitchen. Participants also completed additional measures included as possible predictors of teaching kitchen program engagement. These new measures are detailed below. The order of all survey questions was randomized within each survey.

One important potential predictor of taking a cooking class is awareness of the teaching kitchen program. To measure teaching kitchen program awareness, we asked participants the following question: “Are you familiar with your workplace’s Teaching Kitchen program?” Participants selected either “Yes” or “No”.

Health-related factors were also included as potential predictors of taking classes. These single-item measures indexed different facets of health and wellbeing, especially those that relate to food. Our first measure was overall wellbeing, which we modeled after the commonly used single-item measure of general self-rated health [60]. Participants were asked to rate their overall wellbeing (response options: excellent (4), good (3), fair (2), and poor (1)). Our second measure was healthy eating frequency, in which participants were asked to indicate how often they eat meals they consider to be healthy (response options: all the time (4), most of the time (3), often (2), sometimes (1), never (0)). The third measure was healthy eating prioritization, wherein participants were asked to rate how much they agreed with the phrase “Eating healthy is a priority for me” (response options: strongly agree (4), agree (3), neither agree nor disagree (2), disagree (1), and strongly disagree (0)). Our final health-related factor was home cooking frequency, where participants were asked to indicate how many times they cook from home on an average week by entering a number into an open-response box. 

The final category of potential predictors of taking a cooking class we included were social factors. Classes are a social experience, and as such, social factors may play a role in leading employees either to engage with or avoid these classes. We included two single-item measures: interest in getting to know coworkers (“I am interested in getting to know other employees through shared experiences”) and intimidated to cook around coworkers (“I am intimidated by the idea of cooking around other employees”). Response options for both single-item measures were as follows: strongly agree (4), agree (3), neither agree nor disagree (2), disagree (1), and strongly disagree (0).

#### 2.2.3. Statistical Analysis

All analyses were conducted using R Statistical Analysis Software verion 4.2.3 (R Core Team [2021]. R: A language and environment for statistical computing. R Foundation for Statistical Computing, Vienna, Austria. Available online: https://www.R-project.org/, (accessed on 7 March 2024), all tests were two-tailed, and statistical significance was assigned when the *p* value was less than 0.05. We use three main inferential statistical methods in this work: *t*-tests, correlations, and logistic regression. All key assumptions of *t*-tests (continuous dependent variables; independent observations, normal distribution, and lack of outliers) held for each *t*-test. All key assumptions (continuous independent observations, linear relationships, normal distributions, and lack of outliers) held for each correlation. Finally, all key assumptions for logistic regression (binary dependent variables, independent observations, lack of multicollinearity) held for each logistic regression.

## 3. Results

### 3.1. Study 1

#### 3.1.1. Assessment of the Food Literacy Survey

As a crucial initial step, we evaluated the food literacy surveys completed by the participants. To be effective measures, these surveys need to be both selective (meaning each measure captures something distinct from the others) and reliable (with the items within a measure being internally consistent). Since each survey aims to gauge a positively valenced attitude toward cooking, we anticipated positive correlations among them (indicating that those with higher confidence in the kitchen will likely also score higher in food skills). If these conditions were met—if the measure proved to be selective, reliable, and positively correlated—it would affirm the surveys’ effectiveness in assessing various aspects of food literacy. Our evaluation focused on pre-class surveys, which included responses from 9965 participants.

An exploratory factor analysis using maximum likelihood extraction was conducted to assess whether the above attitude measures captured separate constructs from one another. Promax rotation was used to generate rotated solutions. Extraction yielded four factors with eigenvalues above one, so four factors were retained (see Table 1). The rotated solution showed that the factors corresponded perfectly to the different constructs detailed above: Factor 1 was made up entirely of food skills items, Factor 2 was made up entirely of importance of learning experience items, Factor 3 was made up entirely of ease and pleasure of cooking items, and Factor 4 was made up entirely of confidence in the kitchen items (see Table 2). These results indicate that the survey measures employed successfully captured different constructs from one another, providing evidence that it is appropriate to consider them as distinct variables in our analysis.

Next, we assessed the internal reliability of each of the four food literacy measures by computing Cronbach’s α estimates for each measure. For all four food literacy measures, reliability estimates were very high: importance of the learning experience: 0.91; ease and pleasure of cooking: 0.87; food skills: 0.93; confidence in the kitchen: 0.90. This indicates that each of our food literacy measures were internally consistent.

We also tested whether these metrics would be positively correlated with one another as we would expect from measures about several positively valenced attitudes toward the same topic (cooking). Results showed that, as expected, all measures were significantly positively correlated—see Table 3. 

Taken together, the above assessment of these survey metrics shows that these measures are selective (demonstrated by the factor analysis results), internally reliable (demonstrated by the reliability analyses), and positively associated with one another in line with our predictions (demonstrated by the correlation results). This pattern of results provides evidence that these surveys are effective measures of the different facets of food literacy that we employed in this work, and as such are appropriate for use in assessing correlations between taking classes and attitudes toward cooking.

#### 3.1.2. Descriptive Statistics

We next computed descriptive statistics to better understand the population of employees who take classes (using pre-class surveys; N = 9665). See Table 4 for descriptive statistics for each food literacy measure.

Descriptive results indicated that class attendees reported high levels of importance of the learning experience and ease and pleasure of cooking (both scores had averages well above the midpoint of 2). Confidence in the kitchen was, on average, slightly above the midpoint of the scale, suggesting that class attendees had moderately high confidence in their ability to cook but still had substantial room to grow. Food skills, by contrast, was the only cooking attitude with an average below the scale midpoint, suggesting that class attendees feel they have a great deal to grow when it comes to proficiency in specific food skills.

#### 3.1.3. Assessing Whether Food Literacy Scores Change after Taking Classes

A core question of this research was to assess whether food literacy scores change after employees take classes. For each food literacy question set, we ran paired *t*-tests to examine whether there were significant changes in participants’ scores from their pre-class survey compared to their post-class survey. If participants completed multiple post-class surveys during the study period, their most recent post-class survey was used in this analysis. Of the 720 participants who took both pre- and post-class surveys, 86 were found to have taken them out of order. We excluded these participants from these analyses, resulting in a total of 634 participants. Some participants also failed to complete all the survey items, so the degrees of freedom vary slightly between analyses below. 

Results (visualized in Figure 1) indicate that compared to their pre-class survey, participants showed no significant change in confidence in the kitchen (*t*(621) = −0.72, *p* = 0.475, Cohen’s *d* = −0.01), suggesting that confidence was neither boosted nor diminished after taking classes. We found a modest but significant increase in importance of the learning experience (*t*(631) = 2.07, *p* = 0.039, Cohen’s *d* = 0.07), suggesting that taking classes may be associated with a slight increase in the extent to which participants believe that cooking and eating healthy, sustainable food is important. We found a modest but significant decrease in ease and pleasure of cooking (*t*(633) = −3.46, *p* = 6 × 10^−4^, Cohen’s *d* = −0.11), suggesting that participants may feel that cooking at home is slightly less simple or enjoyable than they did before taking classes (this result is unpacked further below). Finally, we found significant evidence of substantial growth in food skills (*t*(601) = 7.99, *p* = 7 × 10^−15^, Cohen’s *d* = 0.28), suggesting that participants experience a significant increase in their self-rated cooking ability after attending classes.

#### 3.1.4. Post-Hoc Analysis

We wanted to address three questions: (1) What accounts for the modest decrease we observed in ease and pleasure of cooking? (2) For the food literacy measures where we saw growth (importance of the learning experience and food skills), does the amount of growth depend on the number of classes taken? (3) Did those with lower levels of initial food literacy grow more than those with higher levels of food literacy after taking classes?

To better understand the decrease in ease and pleasure of cooking we observed, we ran paired *t*-tests comparing pre- and post-class responses for each of the five items that make up the ease and pleasure of cooking score. Interestingly, results indicated that there was only a significant decrease in one item: “I can fit cooking into my schedule” (pre-class score *M* = 2.97, post-class score *M* = 2.49, *t*(633) = −11.95, *p* < 2 × 10^−16^, Cohen’s *d* = −0.47). All other items showed either no significant change (all *p*s > 0.05) or an increase comparing post-class scores to pre-class scores (“I enjoy eating the food I make”; pre-class score *M* = 3.35, post-class score *M* = 3.42, *t*(633) = 2.30, *p* = 0.022, Cohen’s *d* = 0.09). These results suggest that the main decrease that occurred after taking classes on ease and pleasure of cooking was that class attendees felt less able to find time to cook.

We next assessed whether the significant growth in food skills we observed between pre- and post-class surveys was dependent on the number of classes attendees took between taking their pre- and post-class surveys. For these analyses, we focused only on attendees that took on-site classes, as attendance records were acknowledged to be more accurate for on-site classes (where an instructor can easily mark an attendee as present or not) compared to virtual classes. For this analysis, we were specifically interested in understanding whether a specific number of classes was necessary to see food skills gains. As a result, we did not rely on linear analyses, but instead categorical ones. This approach allowed us to compare, say, growth associated with taking one class to growth associated with taking four classes. We limited our analysis to just those who took four or fewer on-site classes during the study period, as there was a considerable drop-off in the number of people who took more than four classes.

Results indicated that, perhaps surprisingly, the amount of growth on these attitudes did not depend on the number of classes taken. Taking just one class was associated with a significant increase in food skills (*t*(287) = 5.74, *p* = 2 × 10^−8^, Cohen’s *d* = 0.30), and significant increases were observed for taking two, three, and four classes as well (all *p*s < 0.05). While there was a directional increase in the extent to which food skills grew after taking additional classes, no significant differences were found between the amount of growth associated with taking different numbers of classes (all *p*s > 0.05). It should be noted, however, that the sample size decreased as the number of classes taken increased, so our ability to detect significant differences as a function of the number of classes taken was limited (See Figure 2).

As a final post-hoc analysis, we wished to understand whether the amount of growth in food literacy after taking classes depended on where attendees’ food literacy starting point was. It was possible, for instance, that those with lower levels of initial food literacy may have had more room to grow and would therefore grow more than those with higher initial levels of food literacy. Conversely, it was also possible that we would observe a “rich-get-richer” effect, whereby those with higher starting levels of food literacy would be able to take better advantage of their experience with classes to grow more.

Results across every food literacy measure suggested that the former was the case—those with lower levels of starting food literacy grew significantly more than those with higher levels of food literacy initially. The correlations between initial food literacy levels and change in food literacy levels were significantly negative for each metric: ease and pleasure, *r* = −0.440; importance, *r* = −0.565; food skills, *r* = −0.445; confidence, *r* = −0.437; all *p*s < 0.001. See Figure 3.

To summarize our findings from Study 1, we found that our measures of food literacy were selective, reliable, and correlated with one another in ways that suggest that our measures successfully capture different elements of cooking attitudes (in particular, importance, ease and pleasure, food skills, and confidence in the kitchen). Using these measures, we were able to assess whether those who take classes experience significant changes in their food literacy proficiency after completing their classes. The most substantial change we observed was on food skills—scores on food skills were significantly higher after attending classes than before attending them. Moreover, we found that taking even a single class was associated with significant increases in food skills. We also found that those with lower levels of starting food literacy experienced significantly greater increases on all food literacy metrics than those with higher starting levels of food literacy.

### 3.2. Study 2

#### 3.2.1. Descriptive Statistics

Descriptive statistics for all measures can be found in Table 5. In our sample of 272 participants, 84 reported having taken at least one class (30.9%).

#### 3.2.2. Predictors of Taking a Class

We first wished to assess the extent to which different factors were predictive of taking classes. Logistic regression was used for all analyses. The DV for each regression model was the variable class attendance, where a value of 1 indicates that a participant had taken at least one class and a value of 0 indicates that a participant had not taken any classes offered by the teaching kitchen program. The categories of predictors we tested were as follows: awareness of the teaching kitchen program, health-related factors, cooking attitudes, and social factors. We consider each of these factors in turn below. Each continuous predictor was standardized before being entered into the regression model. Our analytic plan was to consider each predictor separately and end by putting all significant predictors into the same regression model to understand whether these predictors account for unique variance in class attendance. 

In our sample of 272 participants, 114 were aware their workplace had a teaching kitchen program (41.9%). Results indicated that program awareness was a strong predictor of class attendance (β(270) = 3.04, *p* < 2 × 10^−16^). Those who were aware of the teaching kitchen program were 8.32 times more likely to take classes as those who were not. In fact, those who were aware the teaching kitchen program existed were more likely than not to have taken a class (63%). These findings suggest that increasing awareness of teaching kitchen programs is very likely to increase the number of people who take classes.

Four single-item measures were used as our health-related factors: overall wellbeing, healthy eating frequency, healthy eating prioritization, and home cooking frequency. Results indicated that none of these factors were predictive of class attendance (all *p*s > 0.05). These results suggest that those who attend classes are not different from those who do not participate with respect to self-assessed wellbeing and healthy eating. 

The cooking attitudes we considered as possible predictors of class attendance were the same as in Study 1: ease and pleasure of cooking, importance of the learning experience, food skills, and confidence in the kitchen. Results indicated that of these cooking attitudes, both ease and pleasure of cooking (β(270) = 0.300, *p* = 0.029) and confidence in the kitchen (β(270) = 0.271, *p* = 0.047) were predictive of taking classes. Neither importance of the learning experience (β(270) = 0.175, *p* = 0.198) nor food skills (β(270) = 0.047, *p* = 0.720) were significantly predictive of class attendance. These results suggest that those that enjoy cooking and feel confident in their cooking abilities are more likely to take classes.

Two single-item social factors were considered as predictors: interest in “Getting to Know Coworkers” and “Intimidated to Cook Around Coworkers”. Results indicated that while interest in “Getting to Know Coworkers” was not a significant predictor of class attendance (β(270) = 0.162, *p* = 0.231), intimidated to “Cook Around Coworkers” was highly predictive as a barrier to class attendance (β(270) = −0.616, *p* = 9 × 10^−5^). These results suggest that feeling intimidated about the prospect of cooking around one’s coworkers may act as an important deterrent to engaging with classes.

The above analyses identified four significant predictors of class attendance: teaching kitchen program awareness, ease and pleasure of cooking, confidence in the kitchen, and feeling intimidated to cook around coworkers. Here, we wished to test the extent to which these factors were predictive of unique variance in class attendance. As a result, we entered each predictor into the same regression model. Results, shown in Table 6, indicate that when accounting for each other factor, only two significant factors remain: teaching kitchen program awareness and feeling intimidated to cook around coworkers. Together, these results suggest that raising awareness and finding ways to combat feelings of intimidation around cooking with coworkers could be two key techniques to increase class attendance with workplace teaching kitchen programs.

#### 3.2.3. Comparing Growth in Food Literacy Measures between Class Attendees and Non-Attendees

The data collected in Study 2 also allowed us to examine a “control group” of employees who had never taken classes and assess whether their cooking attitudes changed over time between their pre- and post-surveys. In Study 1, we found that class participants increased modestly in importance of the learning experience and substantially in food skills. However, in that study alone, we had no comparison group of employees who did not take classes. By comparing these pre- and post-survey results to results from participants who did not take any classes, we can rule out the possible explanation for the growth we saw in class attendees that people simply grow on these metrics over time.

Of the 156 participants who completed the post-survey, here, we focused on the 87 participants who had never taken a class either before the study period or during the study period. Results indicated that these participants did not significantly grow on any of the cooking attitude measures (importance of the learning experience: *t*(86) = −1.05, *p* = 0.298, Cohen’s *d* = −0.10; ease and pleasure of cooking: *t*(86) = 1.23, *p* = 0.220, Cohen’s *d* = 0.06; confidence in the kitchen: *t*(86) = 1.26, *p* = 0.212, Cohen’s d = 0.09; food skills: *t*(86) = 1.48, *p* = 0.143, Cohen’s *d* = 0.078). These results provide important additional context to our findings in Study 1: they suggest that people do not seem to spontaneously change on these cooking attitude metrics over the time interval of a few weeks or months. As a result, we can rule out spontaneous changes over time as a possible explanation of the growth we saw in importance of the learning experience and food skills among those who took classes.

## 4. Discussion

The dual objectives of this research were to assess the potential to increase food literacy in employees after participating in workplace teaching kitchen cooking classes and to identify the key predictors of employee engagement with the program. 

### 4.1. Impact on Measures of Food Literacy

We observed a significant growth in “Food Skills”, which was evident even after a single class, with a trend suggesting continued growth with additional classes. This outcome underscores the program’s pivotal role in successfully fostering skill development among adults. “Food Skills” represent a crucial aspect of food literacy, encompassing elements essential for creating and consuming a balanced meal, including planning, selecting, preparing, and consuming [44,56]. Our findings concur with the existing literature from a variety of countries and population groups [39,50,52,53,61,62,63,64,65,66] and provide new insights that these skills can be built one class at a time.

We found a modest but significant increase in intrinsic motivation for the importance of the learning experience after participants took one or more cooking classes. A plausible explanation may be participating in the class made the value of the experience more salient and tangible to participants. Also, Luo and colleagues found that expertise contributes to perceived benefits and value associated with an activity [67]. Expertise is acquired over time through past consumption, thereby forming a chain linking expertise to the perceived value associated with the activity.

Contrary to our expectations, confidence measures remained unchanged post-teaching kitchen classes, suggesting that while a single session may significantly contribute to skill development, it may not suffice to influence underlying confidence factors. The literature shows measures of confidence have generally increased after comparable nutrition education and cooking interventions in a variety of populations; however, these interventions have exposed participants to at least six classes and lasted between one and six months [39,40,50,52,53,62,63,64,65,66,68]. Confidence is a measure that involves a perception of autonomy, competence, and relatedness and it is crucial for individuals’ ability to make choices and control their lives. Further exploration is warranted into factors influencing confidence in the context of employee teaching kitchen cooking classes. Specifically, investigating whether confidence increases with additional classes as participants gain competence and experience enhanced relatedness within the community. 

An intriguing finding in our study was the statistically significant decrease in “perceived ability to find time to cook” within the class attendees. While seemingly counterintuitive, this shift could stem from a heightened awareness and a more realistic appraisal of the time commitment required for successful cooking. Participants may have gained a more accurate understanding of the time investment associated with preparing meals, reflecting a nuanced understanding of the planning and preparation involved. It is noteworthy that time scarcity, the sense of insufficient time, tends to decrease as individuals become more competent or expert in an activity [69]. Tracking this relationship over multiple classes or an extended period may offer further insights. 

Another noteworthy finding from this work was that those with lower levels of initial food literacy grew significantly more than those with higher levels of initial food literacy after taking teaching kitchen classes. This suggests that teaching kitchen classes may be particularly beneficial for those who have little experience in the kitchen to start with.

In a secondary analysis comparing the food literacy measures of participants who engaged via our virtual platform with those on-site, we found no differences between the responses of the two groups. Our live, hands-on cooking classes were designed with the same pedagogical structure to enhance food literacy and employed the principles of social cognitive theory for an interactive learning experience. This insight is valuable for understanding the scalability and operationalization of this approach. While other literature has shown a positive impact on measures of self-reported food literacy [61,65,66], not all has [70]. Further research is needed to understand how to effectively foster food literacy with programs delivered online or virtually.

Previous intervention studies have generally shown a positive connection between cooking classes and outcomes related to diet, health, and psychosocial wellbeing [36]. However, the specific impact of these classes, as distinct from other elements within these multidisciplinary interventions, remained unclear [71]. Additionally, the optimal exposure required to induce behavioral change was uncertain [71]. Our research contributes clarity to this landscape by demonstrating that teaching kitchen classes, grounded in the principles of behavioral science, crafted with a pedagogy focused on enhancing food literacy and leveraging social cognitive theory, led to a positive increase in food skills and intrinsic motivation among participating employees. Remarkably, these improvements were evident even after a single class and can be expected to grow incrementally with more classes. Furthermore, this experience can be successfully delivered through a virtual platform, unlocking expanded access and reducing operating costs. The potential scalability of a program targeting working-age adults could significantly enhance wellness and reduce the long-term risk of chronic diseases.

### 4.2. Predictors of Employee Participation

The identification of program awareness as the foremost predictor of participation underscores the critical role of communication and marketing strategies for teaching kitchens within the workplace. To maximize engagement, organizations should invest in comprehensive employee campaigns that highlight the benefits of the teaching kitchen program and reach employees across diverse channels. This finding reinforces the notion that effective communication is not just informative but serves as a catalyst for active employee involvement. 

The observed link between prior participation in virtual cooking classes and increased likelihood of joining a workplace class unveils a potential pathway for introducing individuals to the program. This “first experience” or “trial” through the virtual platform can serve as a gateway, sparking interest and confidence. Understanding the specific aspects that resonate with employees in virtual settings can guide the design of introductory experiences, tailoring them to meet the needs and preferences of potential participants. To enhance participation rates, it is crucial to delve into the nuances behind the virtual route into the program. By uncovering what aspects of the program appeal to employees during this initial exposure, organizations can refine program descriptions and scheduling strategies. For instance, emphasizing that classes are suitable for beginners, highlighting the opportunity to share cooking experiences with family, or focusing on creating personalized, delicious meals from home kitchens may be potent messages in attracting diverse groups of employees.

Surprisingly, the importance of the learning experience did not emerge as a predictor of class participation, suggesting that motivation to attend a cooking class may not necessarily stem from a fundamental interest in acquiring skills. While this may not be a predictor for initial class sign-up, it is likely a predictor of repeat sign-up as we saw a significant increase after participants attended a class. 

Contrary to expectations, neither confidence in cooking nor the frequency of current cooking activities emerged as significant predictors. This suggests that the decision to enroll in a cooking class is influenced by factors beyond an individual’s perceived competence or existing cooking habits. Recognizing that this experience is not limited to employees who are already cooking or confident in their cooking skills provides valuable insights into the potential impact of the program for individuals who would benefit from enhanced food literacy. Additionally, this understanding helps provide nuanced insights for tailoring program content and promotional strategies.

Surprisingly, factors traditionally associated with social interaction and health motivation, such as getting to know colleagues or expressing an interest in health-related changes, did not predict participation. This challenges the assumption that communal experiences [72,73,74] or health-conscious attitudes [75] inherently drive engagement and the proposition of a teaching kitchen class is limited to employees with these motivations. 

While not a strong predictor, a weak positive association between food skill scores and enjoyment of cooking did show a connection with enrollment in a class. This suggests that individuals who derive pleasure from the act of cooking might be more inclined to participate, even if their skill level is not a primary driver. This conforms with the field of gastronomy tourism, where pleasure, enjoyment, and hedonism are predictors of participation independently of knowledge or skills [72,74]. Understanding the relationship between enjoyment and participation can inform program design to enhance the overall class experience.

A notable barrier to enrollment was the apprehension and intimidation associated with the prospect of cooking with colleagues. This finding underscores the importance of creating a supportive and inclusive environment. Mitigating feelings of anxiety can be achieved through targeted interventions, such as introductory sessions or team-building activities that foster a sense of camaraderie. Interestingly, we found that feeling more intimidated about cooking around coworkers negatively predicted teaching kitchen engagement even when the classes were only virtual (i.e., participants did not need to interact in a physical space). There are multiple possibilities for why this might be the case. One possibility hinges on the idea that even virtual classes include an element of social interaction. The virtual classes offered encourage attendees to keep their cameras turned on, and as a result attendees can still see their peers as they are cooking. It is possible that even cooking in sight of colleagues virtually is enough to keep those who are intimidated by cooking around their coworkers away. Another possibility is that those who have never engaged with a virtual teaching kitchen class simply do not know how much social interaction occurs at these virtual classes. For instance, they may believe that they might be put on the spot in front of their colleagues. Future work could assess whether completely camera-off virtual classes—wherein the marketing for the class makes it clear there is no social interaction at all—would be enough to encourage those who are intimidated by the prospect of cooking around their coworkers to join for a virtual teaching kitchen class.

Understanding the predictors of participation not only aids in targeting the right audience but also carries implications for the potential impact of the teaching kitchen program on shifting eating habits. As the program aims to influence food choices, the insights gained from this study can inform interventions and initiatives that resonate with the identified participant profile, maximizing the program’s effectiveness in promoting healthier eating habits among employees. 

### 4.3. Limitations and Methodological Considerations 

One notable limitation of this research is the utilization of a convenience sampling method and a single-group prospective study design. This introduces the potential for selection bias, as employees who voluntarily enrolled in teaching kitchen classes may differ systematically from those who did not. To mitigate this concern, we incorporated a quasi-control group of employees who did not participate in teaching kitchen classes, allowing for a preliminary comparison of baseline and post-class metrics.

A further limitation arises from the absence of demographic information collected from participants due to adherence to employee privacy policies. While this safeguards individuals’ privacy, it restricts our ability to analyze potential demographic influences on the outcomes observed. Future research endeavors should consider strategies for ethically collecting demographic data while respecting privacy constraints.

The fact that teaching kitchen classes at this employer were offered free of charge presents a limitation in assessing the impact of price or cost on behavior. Exploring the influence of cost on participation and behavior change would be an insightful avenue for future investigations, allowing for a more comprehensive understanding of the factors affecting engagement in workplace-based teaching kitchen programs.

Acknowledging the workplace-specific nature of our study is crucial. The outcomes may be influenced by the unique characteristics of the workplace environment under examination. Generalizing findings to other organizational contexts should be approached with caution, recognizing the potential variation in workplace cultures, policies, and employee demographics.

### 4.4. Future Research and Recommendations 

To enhance the robustness of our findings, future research should consider a randomized controlled intervention study design to help control for confounding variables and strengthen the causal inference regarding the impact of cooking classes on food literacy.

Delving deeper into the motivational aspects of participants represents a promising avenue for future research. Investigating further the specific factors that drive individuals to enroll in teaching kitchen classes may broaden the appeal to a broader range of participants and understanding how motivation influences sustained behavior change over time would contribute valuable insights to the field. 

Expanding the temporal scope of research by exploring the long-term effects of teaching kitchen interventions on cooking and food behaviors, food choices, and dietary patterns would provide a more comprehensive understanding of the sustained impact of food literacy education in the workplace.

Examining the scalability and sustainability of teaching kitchen programs across diverse workplace settings is essential. Insights into the adaptability of such programs in different organizational contexts would inform the development of strategies for widespread implementation, contributing to organizational wellness initiatives on a broader scale.

Additionally, future work should assess the impact of teaching kitchen programs on important attitudes toward cooking across multiple countries. While the current work had representation from 54 countries, the number of participants within each country did not provide sufficient statistical power to make country-by-country comparisons. Future work conducted with this specific question in mind could employ a research design aimed to specifically understand national differences in the impact of teaching kitchen programs.

## 5. Conclusions

Our study supports the value of workplace teaching kitchens in promoting healthier and more sustainable eating habits among employees. Creating accessible workplace food and nutrition education programs, with a focus on hands-on instruction in teaching kitchens, is an effective strategy to facilitate the shift from processed foods to balanced home-cooked meals. This approach surpasses traditional didactics of nutrition education and dietary guidance by offering an immersive and social learning experience that focuses on building food literacy. To fully realize the potential of teaching kitchens as public health tools, further research is needed to tailor content for diverse workforce settings and assess long-term impacts on eating behaviors. This study underscores the importance of ongoing exploration and refinement of teaching kitchen interventions in fostering a culture of informed and sustainable dietary choices in the workplace.

## Figures and Tables

**Figure 1 nutrients-16-00865-f001:**
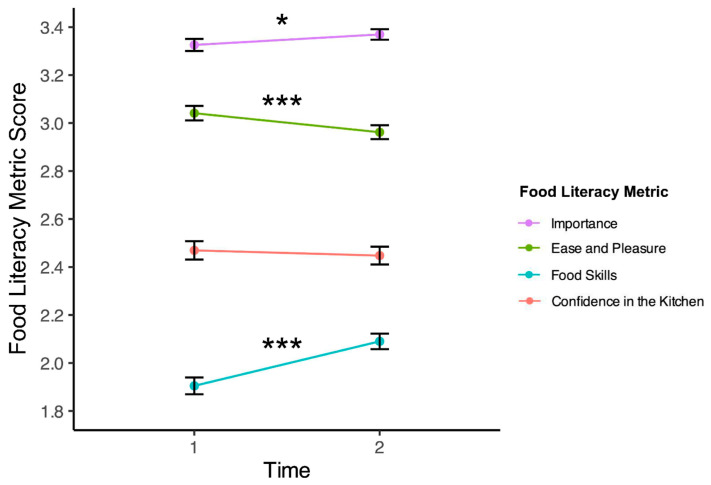
Change in food literacy measures after teaching kitchen classes. This figure shows the values of each food literacy measure by time (1 = pre-survey; 2 = post-survey). Error bars represent standard errors. One star (*) indicates a significant difference between pre- and post-survey values at *p* < 0.05, and three stars (***) indicates a significant difference between pre- and post-survey values at *p* < 0.001 using paired *t*-tests.

**Figure 2 nutrients-16-00865-f002:**
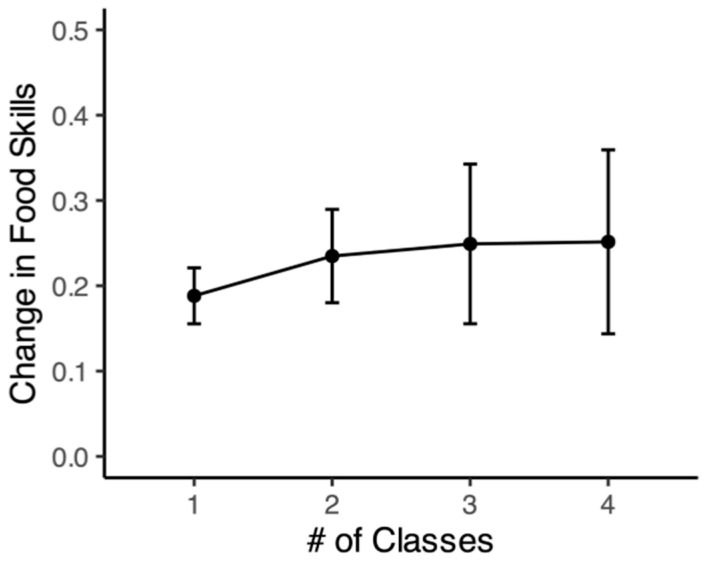
Change in food skills by number of teaching kitchen classes taken. This figure shows the change in food skills by the number (#) of on-site classes taken in between the pre- and post-surveys. Error bars represent standard errors.

**Figure 3 nutrients-16-00865-f003:**
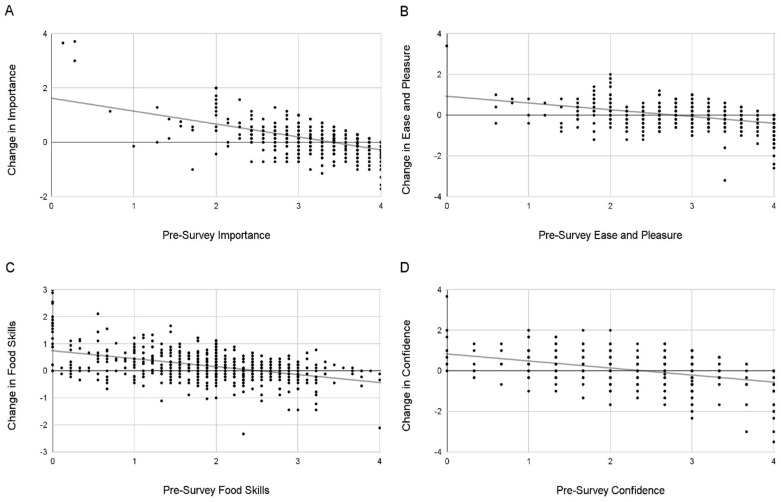
Food literacy changes as a function of initial food literacy levels. This figure shows scatterplots of food literacy scores on the pre-survey on the *x*-axis and the change in food literacy scores on the *y*-axis for each food literacy measure. Lines of best fit are shown. (**A**) Importance of cooking (**B**) Ease and pleasure of cooking (**C**) Food skills (**D**) Confidence in the kitchen.

**Table 1 nutrients-16-00865-t001:** Food literacy survey factor analysis eigenvalues.

Factor	Eigenvalue
1	8.93
2	4.30
3	1.67
4	1.59
5	0.75

Note. Eigenvalues for the top five factors yielded by maximum likelihood extraction of responses to food literacy items. Factors 1–4 were retained. The retained factors corresponded to the following constructs: 1—food skills; 2—importance of the learning experience; 3—ease and pleasure of cooking; 4—confidence in the kitchen.

**Table 2 nutrients-16-00865-t002:** Food literacy survey rotated factor loadings.

Item	Factor 1	Factor 2	Factor 3	Factor 4
Food skills 1	0.69			
Food skills 2	0.76			
Food skills 3	0.75			
Food skills 4	0.85			
Food skills 5	0.82			
Food skills 6	0.82			
Food skills 7	0.74			
Food skills 8	0.76			
Food skills 9	0.77			
Importance 1		0.54		
Importance 2		0.79		
Importance 3		0.87		
Importance 4		0.76		
Importance 5		0.66		
Importance 6		0.80		
Importance 7		0.84		
Ease and pleasure 1			0.67	
Ease and pleasure 2			0.82	
Ease and pleasure 3			0.81	
Ease and pleasure 4			0.69	
Ease and pleasure 5			0.75	
Confidence 1				0.87
Confidence 2				0.91
Confidence 3				0.63

Note. Rotated factor loadings for the four-factor solution for all 24 items. Loadings with an absolute value below 0.3 are suppressed.

**Table 3 nutrients-16-00865-t003:** Correlations between food literacy measures.

Measure	Importance	Ease and Pleasure	Food Skills	Confidence in the Kitchen
Importance	1			
Ease and pleasure	0.444	1		
Food skills	0.157	0.490	1	
Confidence in the kitchen	0.237	0.468	0.546	1

Note. Correlations between all food literacy measures. All correlations are significant at *p* < 2 × 10^−16^. *df* = 9663.

**Table 4 nutrients-16-00865-t004:** Descriptive statistics of food literacy measures.

Measure	Mean	SD
Importance	3.28	0.70
Ease and pleasure	2.95	0.80
Food skills	1.74	0.92
Confidence in the kitchen	2.41	0.99

Note. Descriptive statistics of food literacy measures. All measures had a possible range of 0 to 4.

**Table 5 nutrients-16-00865-t005:** Descriptive statistics of predictors of teaching kitchen engagement.

Measure	Mean	SD
Importance	3.17	0.61
Ease and pleasure	2.60	0.80
Food skills	1.98	1.02
Confidence in the kitchen	2.42	1.04
Overall wellbeing	2.05	0.72
Healthy eating frequency	2.64	0.80
Healthy eating prioritization	1.74	0.75
Home cooking frequency	3.76	2.28
Getting to know coworkers	1.90	0.84
Intimidated to cook around coworkers	3.69	1.10

Note. Descriptive statistics of all continuous candidate predictors of teaching kitchen engagement are shown.

**Table 6 nutrients-16-00865-t006:** Logistic regression model predicting TK engagement.

Measure	β	*SE*	*p*
Teaching kitchen program awareness	3.040	0.371	3.00 × 10^−16^
Ease and pleasure	0.332	0.196	0.090
Confidence in the kitchen	0.030	0.198	0.881
Intimidated to cook around coworkers	−0.415	0.193	0.031

Note. Logistic regression model results testing for unique prediction of variance in teaching kitchen engagement. *df* = 267. All continuous variables are standardized.

## Data Availability

Data available on request from the corresponding author, due to restrictions on company confidentiality.

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
