# Peer review of "Cultivating Healthier Habits: The Impact of Workplace Teaching Kitchens on Employee Food Literacy"

_nutrients, 2024, doi:10.3390/nu16060865_

Round 1

Reviewer 1 Report

Comments and Suggestions for Authors

The paper entitled Cultivating Healthier Habits: The Impact of Workplace Teaching Kitchens on Employee Food Literacy’ aimed to evaluate the impact of a workplace teaching kitchen program on measures of food literacy in employees and to identify the key predictors of employee engagement with the teaching kitchen program. The manuscript may be interesting to a wide audience, nevertheless, it should be improved, specifically by ‘adding more scientific context’.  

Detailed remarks/comments given below should be addressed by authors.

Materials and Methods:

Page 3: please add some details about onsite and virtual classes – any differences? The numbers of participants in sessions, the duration, etc.

Page 4: Food Skills – it’s not clear how were the points counted as in the supplementary material the points are different (from 0 to 5). Moreover, if that scale had only integer values why it is not shown in Supplementary material? ‘Slider’ and the answer options suggest different data values.

Some parts of data presented in Results should be moved to Materials and Methods section.

Please add the section Statistical analysis and give the details about ALL assumptions, tests, etc. used for data analyses in the whole study.

Results

The text should be retyped to be more concrete (and less wordy), without own interpretation/ comments which should appear in Discussion section.

Tab. 1: add to Note the meanings/names of factors.

Fig. 1: explain DV, put the legend in right order (the same as in tables), improve its quality.

Why did you use the parametric test when such data require non-parametric ones, e.g. Wilcoxon test?

Results/Discussion: ‘Intimidated to Cook Around Coworkers’ was the significant predictor in the regression model; what about the virtual classes which do not bring such negative emotions?

Discussion

Some information can be added – what about the role of education level in food literacy? Although you did not collect such data, do you have any knowledge about the participants’ education? They were employees at an American multinational technology company so probably they had to meet some ‘standards’?

Supplementary material – please add the names of measures.

Some technical/editorial errors that appear in manuscript:

-  references in the text

-   some p values in the text, tables, notes

- lack of DOI in the list of References; some data are missing, e.g. # 12 - the year of publication

Author Response

Thank you very much for taking the time to review this manuscript. Please find the detailed responses below and the corresponding revisions and corrections are highlighted in red typeface in the re-submitted files.

Comment 1.0

The paper entitled Cultivating Healthier Habits: The Impact of Workplace Teaching Kitchens on Employee Food Literacy’ aimed to evaluate the impact of a workplace teaching kitchen program on measures of food literacy in employees and to identify the key predictors of employee engagement with the teaching kitchen program. The manuscript may be interesting to a wide audience, nevertheless, it should be improved, specifically by ‘adding more scientific context’. 

Response 1.0

We appreciate the comment that this work might be of interest to a wide audience and thank the reviewer for their helpful comments, which we believe have contributed to an improved manuscript.

Comment 1.1

Materials and Methods:

Page 3: please add some details about onsite and virtual classes – any differences? The numbers of participants in sessions, the duration, etc.

Response 1.1

In response to the reviewer's comment, we clarified the structure and format for the onsite and virtual online classes on page 3.

The intervention for this research was consumption of one or more classes from the established workplace Teaching Kitchen program at a multinational tech company. Teaching Kitchens are co-located in company offices available in 38 cities worldwide and globally through a virtual online platform spanning all 54 countries the tech company has employees. The program offered is the same whether accessed through an onsite teaching kitchen  or online; including the class format, duration, audio visual tools, curriculum, syllabus and teaching pedagogy. The latter are described in more detail in the next paragraph. The Teaching Kitchen spaces and program are based on the concept described by Eisenberg and colleagues 46

Comment 1.2

Page 4: Food Skills – it’s not clear how were the points counted as in the supplementary material the points are different (from 0 to 5). Moreover, if that scale had only integer values why it is not shown in Supplementary material? ‘Slider’ and the answer options suggest different data values.

Response 1.2

We thank the reviewer for pointing this out.  We erroneously included an old version of the Food Skills measure in the Supplementary Materials. We have also clarified how this scale worked and how we arrived at the score. See page 4.

Food Skills refers to self-rated ability on several key cooking skills. Participants were instructed to “Use the slider to tell us how you would rate your food skills today” and were shown a series of 9 items. Example items include “Plan meals ahead of time”, “Knife skills (chopping, slicing, dicing…)” and “adapt recipes”. Ratings were made on a slider from a frowny face (1) to a smiley face (5), and only integers were selectable. Food Skills scores for each participant were computed by taking the average of the 9 items. We subtracted 1 from this average to arrive at a possible range of Food Skills scores from 0 to 4, which is the same range as the other food literacy measures. Cronbach’s for this measure was 0.93.

Comment 1.3

Some parts of data presented in Results should be moved to Materials and Methods section.

Response 1.3

We are unsure which parts of the data the reviewer is referring to. If the reviewer has the factor analysis results in mind, we opted to include those in the Results section because this line of analysis was deemed too substantial to include in the Materials and Methods section. This analysis of the measures we used was crucial to enable any of the other inferences we wished to make in the paper based on the food literacy survey, and as such we believe it is worthy of inclusion in the Results section.

Comment 1.4

Please add the section Statistical analysis and give the details about ALL assumptions, tests, etc. used for data analyses in the whole study.

Response 1.4

We thank the reviewer for this suggestion and have added this section at the end of the Materials and Methods section on page 6:

2.2.3. Statistical analysis

All analyses were two-tailed. We use three main inferential statistical methods in this work: t-tests, correlations, and logistic regression. All key assumptions of t-tests (continuous dependent variables; independent observations, normal distribution, and lack of outliers) held for each t-test. All key assumptions (continuous independent observations, linear relationships, normal distributions, and lack of outliers) held for each correlation. Finally, all key assumptions for logistic regression (binary dependent variables, independent observations, lack of multicollinearity) held for each logistic regression.

Comment 1.5

Results

The text should be retyped to be more concrete (and less wordy), without own interpretation/ comments which should appear in Discussion section.

Response 1.5

We appreciate this comment. It is our hope that this manuscript is read not only by researchers, but also by practitioners who have an interest in teaching kitchens. As we anticipate this work being shared with a non-research audience, we made the decision to use this style in the Results section so that both researchers and non-researchers alike can follow the logic of our analyses. These non-research readers may not understand every word of the Results section, but it is our hope that using this style of briefly explaining the findings as we go can help them to understand why we ran the tests we ran. We also note that we are careful (both in the Results section and in the Discussion section) that the inferences we draw do not go beyond the data, and as a result we believe our style in the Results section is appropriate. Finally, we do reserve our more detailed unpacking of what the results mean for the Discussion section.

Comment 1.6

Tab. 1: add to Note the meanings/names of factors.

Response 1.6

We thank the reviewer for this comment and have done as the reviewer suggests (see page 7):

Note. Eigenvalues for the top five factors yielded by maximum likelihood extraction of responses to food literacy items. Factors 1-4 were retained. The retained factors corresponded to the following constructs: 1 - Food Skills; 2 - Importance of the Learning Experience; 3 - Ease and Pleasure of Cooking; 4 - Confidence in the Kitchen.

Comment 1.7

Fig. 1: explain DV, put the legend in right order (the same as in tables), improve its quality. 

Response 1.7

We thank the reviewer for these suggestions. We have made the suggested revisions to this figure on page 10 and believe this makes these findings clearer to the reader.

Comment 1.8

Why did you use the parametric test when such data require non-parametric ones, e.g. Wilcoxon test?

Response 1.8

We are not sure which analysis the reviewer has in mind here. However, we believe a parametric approach is appropriate for the analyses used in this work as the overall sample size tended to be fairly large and the dependent variables used in our analyses did not deviate substantially from a normal distribution.  

Comment 1.9

Results/Discussion: ‘Intimidated to Cook Around Coworkers’ was the significant predictor in the regression model; what about the virtual classes which do not bring such negative emotions?

Response 1.9

We thank the reviewer for raising this question, as this is indeed an interesting point of discussion.

We believe that one explanation for why feeling intimidated to cook around coworkers would predict avoidance of even virtual classes hinges on the idea that even these virtual classes involve social interaction. The virtual classes are typically conducted in a video chat style where all attendees are encouraged to have their cameras turned on. Thus, there is still an important social element even to these virtual classes. Moreover, those who have never taken a virtual teaching kitchen class likely do not know what level of social interaction they would have during a virtual class. As a result, those who are intimidated by the idea of cooking around their coworkers may choose to avoid these classes because they fear that they might be put on the spot around their coworkers in that virtual environment. An interesting line of future work could test whether making it clear that there will be no social element to virtual teaching kitchen classes (i.e., all cameras except the instructors are turned off, and attendees just need to follow along in their own kitchens) would encourage those who say they are intimidated by cooking around their coworkers to engage with the teaching kitchen program. We have included a discussion along these lines in the Discussion on page. 17:

A notable barrier to enrolment was the apprehension and intimidation associated with the prospect of cooking with colleagues.  This finding underscores the importance of creating a supportive and inclusive environment. Mitigating feelings of anxiety can be achieved through targeted interventions, such as introductory sessions or team-building activities that foster a sense of camaraderie. Interestingly, we found that feeling more intimidated about cooking around coworkers negatively predicted teaching kitchen engagement even when the classes were only virtual (i.e., participants did not need to interact in a physical space). There are multiple possibilities for why this might be the case. One possibility hinges on the idea that even virtual classes include an element of social interaction. The virtual classes offered encourage attendees to keep their cameras turned on, and as a result attendees can still see their peers as they are cooking. It is possible that even cooking in sight of colleagues virtually is enough to keep those who are intimidated by cooking around their coworkers away. Another possibility is that those who have never engaged with a virtual teaching kitchen class simply do not know how much social interaction occurs at these virtual classes. For instance, they may believe that they might be put on the spot in front of their colleagues. Future work could assess whether completely camera-off virtual classes  – wherein the marketing for the class makes it clear there is no social interaction at all – would be enough to encourage those who are intimidated by the prospect of cooking around their coworkers to join for a virtual teaching kitchen class. 

Comment 1.10

Discussion

Some information can be added – what about the role of education level in food literacy? Although you did not collect such data, do you have any knowledge about the participants’ education? They were employees at an American multinational technology company so probably they had to meet some ‘standards’?

Response 1.10

We thank the reviewer for suggesting this approach. We have added some details to the Introduction on page 3 based on the general information we have on the employees at the company where the research was conducted. 

Study participants were employees at an American multinational technology company who completed classes from March 1st, 2023 to August 17th, 2023. Data collection included pre- and post-class surveys. All class attendees were required to complete a pre-class survey as a prerequisite before taking their first class. In this period, a total of 9,665 employees completed the mandatory pre-class survey. After each class attendees took during this period, they were offered the opportunity to complete an optional post-class survey. During the study period, 720 employees completed at least one optional post-class survey. Demographic information was not collected to adhere to employee privacy policies. Externally published data on the company employees reported  66.8% of employees have a Bachelor's degree and 14.5% have a Masters degree; 59% are aged between 20 - 30 years and 19% between 30 - 40 years; 68.5% of employees are male; 49.7% are White, 18.2% are Asian, 18.28% are Hispanic or Latino, 8.2% are Black or African American (Google Diversity Annual Report 2023, https://about.google/belonging/diversity-annual-report/2023/)

Comment 1.11

Supplementary material – please add the names of measures.

Response 1. 11

We thank the reviewer and have added the names of the measures to the Supplementary material.

Comment 1.12

Some technical/editorial errors that appear in manuscript:

- references in the text

- some p values in the text, tables, notes

- lack of DOI in the list of References; some data are missing, e.g.

# 12 - the year of publication

Response 1. 12

We are deeply appreciative of the reviewer identifying these errors in the manuscript. We have corrected them.

Reviewer 2 Report

Comments and Suggestions for Authors

This is a valuable paper that evaluates the impact of a workplace teaching kitchen program on food literacy and identifies the predictor of employee engagement.

It is also informative to note that program awareness was important and that virtual classes were related to participation.

The following are comments that could be addressed to improve the manuscript.

Comments: 

1.     How did the contents of the program vary from class to class? In addition, were they standardized for different workplaces in different countries? 

2.     In both Study 1 and Study 2, were there any differences in the results across countries or by the number of times employees attended the classes? 

3.     In Study 1, were the scores of the pre-survey taken into account for the changes in food literacy components? Was there any difference between employees who had initially good food literacy and those who did not?

Author Response

Thank you very much for taking the time to review this manuscript. Please find the detailed responses below and the corresponding revisions and corrections are highlighted in red typeface in the re-submitted files.

Comment 2.0

This is a valuable paper that evaluates the impact of a workplace teaching kitchen program on food literacy and identifies the predictor of employee engagement. It is also informative to note that program awareness was important and that virtual classes were related to participation. The following are comments that could be addressed to improve the manuscript.

Response 2.0

We appreciate the positive assessment of the paper. We also thank the reviewer for their thoughtful comments below. Addressing them has made for a stronger, more informative paper.

Comment 2.1

  1. How did the contents of the program vary from class to class? In addition, were they standardized for different workplaces in different countries?

Response 2.1

The revised section of the manuscript gives more details on the program's curriculum design. We specify that while the core educational framework—encompassing menu planning, ingredient selection, preparation minimizing food waste and healthy techniques, and mindful consumption practices—is uniformly applied, the program is tailored to accommodate the cultural and regional nuances of diverse global workplaces. This customization ensures cultural relevance and practical applicability of the cooking practices shared by the food educators. See the excerpt from pages 3-4 below:

The intervention for this research was consumption of one or more classes from the established workplace Teaching Kitchen program at a multinational tech company. Teaching Kitchens are co-located in company offices available in 39 cities worldwide and globally through a virtual online platform spanning all 54 countries the tech company has employees located. The program offered is the same whether accessed through an onsite teaching kitchen  or online; including the class format, duration, audio visual tools, curriculum, syllabus and teaching pedagogy. The latter are described in more detail in the next paragraph. The Teaching Kitchen spaces and program are based on the concept described by Eisenberg and colleagues 46

The Teaching Kitchen program uses a centralized design approach that is customized to local ingredients and cultural preferences by the local teachers. The program is grounded in a behavior-centered design strategy that incorporates key drivers of behavior change, that provides participants the opportunity to perform and practice the desired behavior, fostering intrinsic motivation, and building capability for successful action 54. In this case the desired behavior is to select and cook with wholesome, unprocessed foods to create delicious nutritious meals and snacks personalized for the participants preferences and lifestyle. The Teaching Kitchen program curriculum integrates principles advocating for healthy and sustainable eating, aligning with the Culinary Institute of America's Healthy & Sustainable Menu Principles 55 and contemporary sustainable dietary guidelines 11,14–17 The pedagogical structure is based on a food literacy framework 44,45,56,57. Sessions focus on planning, ingredient selection, preparation, cooking, serving, and eating, emphasizing skill-building, adaptability, and decision-making applicable in diverse food environments 27,42. Fundamental skills such as mise en place, knife proficiency, ingredient ratios, and flavor balancing are integral aspects of each class. The syllabus is tailored to local interests, culture, and environment, covering seasonal and local foods, global cuisines, cultural favorites, and celebratory dishes. Classes are led by trained culinary professionals proficient in the program's pedagogy. They employ participant engagement strategies following the principles of social cognitive theory, fostering a hands-on group learning environment that integrates personal factors, environmental influences, and behavior 58.

Comment 2.2

  1. In both Study 1 and Study 2, were there any differences in the results across countries or by the number of times employees attended the classes?

Response 2.2

We thank the reviewer for raising this question. With regard to different results across countries, we agree that this is an interesting question. Because we had so many different countries represented in both samples (54 countries), the number of participants in each country was very small. As a result, we did not have adequate statistical power to compare results as a function of which country the participant was in. This is an interesting direction for future work, however, and we have added a brief discussion of this research question in the Discussion on page 18:

Additionally, future work should assess the impact of teaching kitchen programs on important attitudes toward cooking across multiple countries. While the current work had representation from 54 countries, the number of participants within each country did not provide sufficient statistical power to make country-by-country comparisons. Future work conducted with this specific question in mind could employ a research design aimed to specifically understand national differences in the impact of teaching kitchen programs.

Regarding the reviewer’s second question about differences in results by the number of times employees attended classes, we agree this is also an interesting question. In Study 1, we analyze the effect of the number of teaching kitchen classes taken on growth in Food Skills, and we find no significant differences in growth in Food Skills as a function of the number of classes taken. In Study 2, we solely examined growth in cooking attitudes in those who reported never having taken teaching kitchen classes either before the pre-survey or between the pre- and post-surveys and found no significant growth on any metrics.

Comment 2.3

  1. In Study 1, were the scores of the pre-survey taken into account for the changes in food literacy components? Was there any difference between employees who had initially good food literacy and those who did not?

Response 2.3

We thank the reviewer for this insightful question. In the original version of the manuscript, we did not examine this question. However, whether those who begin with lower levels of food literacy change more or less than those with high levels of food literacy after taking classes is of substantial theoretical and practical interest in this space. We therefore tested this, and found consistent evidence that those with lower levels of initial food literacy grow significantly more than those with high levels of food literacy across each of our food literacy levels. We have made several additions to the manuscript to address this point.

We set this up as a key post-hoc question in Study 1 on page 10:

We wanted to address three questions: 1) What accounts for the modest decrease we observed in Ease and Pleasure of Cooking? 2) For the food literacy measures where we saw growth on (Importance of the Learning Experience and Food Skills), does the amount of growth depend on the number of classes taken? 3) Did those with lower levels of initial food literacy grow more than those with higher levels of food literacy after taking  classes?

We include multiple paragraphs and a new Figure dedicated to the Results of this analysis on pages 11 - 12:

As a final post-hoc analysis, we wished to understand whether the amount of growth in food literacy after taking classes depended on where attendees' food literacy starting point was. It was possible, for instance, that those with lower levels of initial food literacy may have had more room to grow and would therefore grow more than those with higher initial levels of food literacy. Conversely, it was also possible that we would observe a “rich-get-richer” effect, whereby those with higher starting levels of food literacy would be able to take better advantage of their experience with classes to grow more.

Results across every food literacy measure suggested that the former was the case - those with lower levels of starting food literacy grew significantly more than those with higher levels of food literacy initially. The correlations between initial food literacy levels and change in food literacy levels were significantly negative for each metric: Ease and Pleasure, r = -.440; Importance, r = -.565; Food Skills, r = -.445; Confidence, r = -.437; all ps < .001. See Figure 3.

To summarize our findings from Study 1, we found that our measures of food literacy were selective, reliable, and correlated with one another in ways that suggest that our measures successfully capture different elements of cooking attitudes (in particular, Importance, Ease and Pleasure, Food Skills, and Confidence in the Kitchen). Using these measures, we were able to assess whether those who take classes experience significant changes in their food literacy proficiency after completing their classes. The most substantial change we observed was on Food Skills - scores on Food Skills were significantly higher after attending classes than before attending them. Moreover, we found that taking even a single class was associated with significant increases in Food Skills. We also found that those with lower levels of starting food literacy experienced significantly greater increases on all food literacy metrics than those with higher starting levels of food literacy.

And we include a brief paragraph discussing this finding in the Discussion on page 15:

Another noteworthy finding from this work was that those with lower levels of initial food literacy grew significantly more than those with higher levels of initial food literacy after taking teaching kitchen classes. This suggests that teaching kitchen classes may be particularly beneficial for those who have little experience in the kitchen to start with. 

We thank the reviewer for raising this question, as we believe it significantly adds to the insights this research is able to offer the field.

Round 2

Reviewer 1 Report

Comments and Suggestions for Authors

Statistical analysis: please add the name of statistical program used for analysis and p-value considered to be significant.

Author Response

We thank the reviewer for again so carefully reviewing our updated manuscript. We agree the reference to the statistical package used and the level of statistical significance used throughout this manuscript was missing.

Comment 1.0.

Statistical analysis: please add the name of statistical program used for analysis and p-value considered to be significant.

Response 1.0.

We have updated this section in the Methods section on page 6. Please see the updated text (highlighted in yellow). 

All analyses were conducted using the R Statistical Analysis Software (R Core Team [2021]. R: A language and environment for statistical computing. R Foundation for Statistical Computing, Vienna, Austria. URL https://www.R-project.org/), all tests were two-tailed, and statistical significance was assigned when the p value was less than 0.05

Reviewer 2 Report

Comments and Suggestions for Authors

Thank you for responding and addressing all comments.

Author Response

Thank you for reviewing our updated manuscript. We are sincerely grateful to the insightful question you raised last time that encouraged us to do further analysis of the data. We believe this has really enhanced the manuscript.